# Acute effects of commercial group exercise classes on arterial stiffness and cardiovagal modulation in healthy young and middle-aged adults: A crossover randomized trial

Xavier Melo[1]* Adma Lopes[2,3], Raquel Coelho[2,3], Bruno Simão[2,3], Isabel Oliveira[4], João L. Marôco[5], Sérgio Laranjo[6,7], Bo Fernhall[5], Helena Santa-Clara[2,3]

1 Centro de Investigação Interdisciplinar Egas Moniz (CiiEM), Egas Moniz School of Health and Science, Caparica, Almada, Portugal, 2 Centro Interdisciplinar de Estudo da Performance Humana, Faculdade de Motricidade Humana – Universidade de Lisboa, Oeiras, Portugal, 3 Ginásio Clube Português, Research & Development Department, GCP Lab, Lisboa, Portugal, 4 Faculdade de Ciências da Saúde e do Desporto, Universidade Europeia, Lisboa, Portugal, 5 Exercise and Health Sciences Department, University of Massachusetts Boston, Boston, Massachusetts, United States, 6 Department of Physiology, NOVA Medical School, Faculdade de Ciências Médicas da Universidade Nova de Lisboa, Lisboa, Portugal, 7 Comprehensive Health Research Center, Universidade NOVA de Lisboa, Lisboa, Portugal

* xmelo@egasmoniz.edu.pt

## Abstract

### Background

Arterial stiffness and cardiac autonomic function are crucial indicators of cardiovascular health. Acute exercise and age impact these parameters, but research often focuses on specific exercise activities, lacking ecological validity.

### Methods

We examined the acute effects of commercially available group fitness classes (indoor cycling, resistance training, combined exercise) on arterial stiffness and vagal-related heart rate variability (HRV) indices in twelve young and twelve middle-aged adults. Participants attended four sessions, including exercise and control conditions, with measurements taken at rest and during recovery.

### Results

Middle-aged, but not young adults, showed reductions in central and peripheral systolic blood pressure 20-min into recovery across all exercise modalities (range: -7 to -8 mmHg $p < 0.05$). However, arterial stiffness remained unchanged. Similarly, vagal-related HRV indices (range: -0.51 to -0.90 ms, $p < 0.05$) and BRS (-4.03, $p < 0.05$) were reduced immediately after exercise, with differences persisting 30 min into recovery only after indoor cycling. Resistance and combined exercise elicited similar cardiovagal modulation and delayed baroreflex sensitivity recovery to cycling exercise, despite higher energy expenditure during indoor cycling (+87 to +129 kcal, $p < 0.05$).

**Data availability statement:** Data will be available as supplemental file

**Funding:** The authors thank Foundation for Science and Technology (FCT)/Portuguese Ministry of Science, Technology, and Higher Education (MCTES) for the financial support to CiiEM (10.54499/UIDB/04585/2020) through national funds. CiiEM has provided support through Project 10.54499/UIDB/04585/2020, funded by FCT.

**Competing interests:** The authors have declared that no competing interests exist.

## Conclusion

Acute group fitness classes induce age-dependent alterations in blood pressure, but not in arterial stiffness or cardiovagal modulation. While the overall cardiovascular effects were generally consistent, differences in autonomic recovery were observed between exercise modes, with prolonged effects seen after indoor cycling. This suggests that exercise prescription should consider both age and exercise modality, as well as recovery time. The findings also emphasize the importance of ecological validity in exercise interventions, highlighting that acute effects on cardiovascular health in real-world settings may differ from those observed in controlled laboratory environments (ID: NCT06616428).

## Introduction

The physiological importance of large artery distensibility in cardiovascular function is well-established. Distensible large arteries contribute to reduced impedance to systolic ejection, slower pulse wave velocity (PWV), and delayed return of reflected pressure waves after aortic valve closure. This dynamic facilitates coronary perfusion during diastole [1]. Conversely, central artery stiffening leads to elevated systolic blood pressure (SBP), lowered diastolic blood pressure (DBP), increased left ventricular afterload, and altered coronary artery perfusion [2]. These changes may induce left ventricular hypertrophy [3], and fatigue in the arterial wall [4], elevating the risk of cardiovascular events and all-cause mortality [5,6].

The autonomic nervous system plays a pivotal role in blood pressure regulation, and its dysregulation is associated with cardiovascular risk [7]. Indices of cardiac autonomic function, such as heart rate recovery (HRR), heart rate variability (HRV), and baroreflex sensitivity (BRS) are closely correlate with central and peripheral blood pressure [8–10]. Post-maximal exercise HRR and HRV are powerful independent predictors of mortality in healthy and clinical populations [11–14]. Reduced BRS, in particular, is a significant marker of autonomic dysfunction and is associated with increased cardiovascular disease risk [15], and has prognostic value for predicting cardiovascular events [16].

Arterial stiffness and cardiac autonomic function indices also predict therapeutic success [17,18]. Cardiovascular medications, including beta-blockers, are known to reduce central PWV and wave reflection, although these changes are primarily driven by reductions in blood pressure, rather than direct effects on arterial stiffness itself [19]. Similarly, other therapeutics, such as exercise, reduce central PWV and wave reflection following a single exercise bout [20,21], impacting cardiac autonomic function indices [22–25]. This research on the "physiology of recovery" has the potential to refine exercise recommendations for both health and performance [26]. Notably, adaptations to physical training may arise not only from the cumulative effects of exercise bouts [27] but also from exercise-induced changes influenced by the mode of exercise [20,21,28,29]. Aerobic exercise has been shown to significantly reduce arterial stiffness, and this effect is amplified with higher exercise intensity, particularly in participants with greater arterial stiffness at baseline [29]. Comparable outcomes were also observed in stretching exercises, but not in resistance training [20]. However, it is important to note that these findings are not universally applicable [29].

Few studies have compared HRR and HRV responses during different exercise modalities. Post-exercise parasympathetic reactivation seems to be determined by the amount of muscle mass and/or energy expenditure and intensity [30,31]. Importantly, these responses are not solely dependent on exercise mode; age, duration, chronic conditions, and the evaluation time point also exert notable influences on the time course changes in arterial stiffness and cardiovagal modulation [32–34].

While these studies focused specifically on aerobic or resistance exercise, it is crucial to recognize that typical sessions aligned with cardiovascular health guidelines encompass a broader spectrum of activities, exemplified by the dynamic nature of group fitness classes [35]. Thus, an evident drawback of the available evidence is the reduced ecological validity [36,37]. Participation in group fitness classes is an increasingly common method by which people meet physical activity recommendations for enhancing and maintaining cardiovascular fitness [35]. The popularity of group fitness classes with the public has been demonstrated for more than two decades, likely attributed to the social and non-competitive class environment. A diverse range of group fitness classes is now available in both commercial and community fitness centers, incorporating various exercise modalities such as stationary cycling, step classes, group resistance exercise, Pilates, and aerobic dance. These classes use a combination of music and instructor-choreographed routines designed to accommodate varying fitness levels. However, despite the high participation rates in group fitness classes, there is currently no available data that describes and/or compares the vascular and autonomic responses and/or metabolic costs of different group fitness classes. Additionally, it remains unclear to what extent participation in such classes impact vascular and autonomic function, and whether there are periods of significant concern arising from transient changes in arterial stiffness or autonomic function immediately following exercise. Thus, this study aimed to assess and compare the acute effects of three commercially available group exercise classes on the recovery time course of indices of arterial stiffness and cardiovagal modulation in both young and middle-aged adults.

## Methodology

### Participants

Twelve young adults aged between 21-34 years old and 12 middle-aged adults aged $\geq 48$ years participated in this study (13 males and 11 females). The flow of participants through the study is presented in Fig 1. Recruitment for the study began on January 1, 2020, and concluded on September 24, 2021. All participants were outwardly active, as assessed by the International Physical Activity Questionnaire (IPAQ), with some experience in both aerobic and resistance exercise ($\sim$3–4 times/week, >3 months). All participants were considered healthy or perceived to be healthy based on the sport's medical examination or the preparticipation screening process, Physical Activity Readiness Questionnaire for Everyone (PARQ+). Exclusion criteria included any form of cardiovascular disease, more than one cardiovascular disease risk factor, resting hypertension (SBP >140 mmHg, DBP > 90 mmHg), any prescription medication use that may influence vascular and autonomic response to exercise, being an athlete, and currently smoking. Prior to the first evaluation day, all participants provided written, informed consent following the approval by the Ethical Committee of the Faculty of Human Kinetics – University of Lisbon (opinion: 35/2019, date: 17th of december 2019).

### Study design

The study was designed as a parallel group randomized, cross-over, repeated-measures trial. All participants were randomly assigned by the principal investigator to 1 of 4 experimental conditions using a randomized block scheme (https://www.randomizer.org/). Participants completed 4 separate intervention sessions, each consisting of an initial rest, a group fitness class of either indoor cycling, resistance training, combined exercise training, or no exercise (CON), followed by a recovery period. A minimum of 72 hours between sessions was ensured. Body composition and cardiorespiratory fitness for each participant were evaluated before and after the CON session, respectively (secondary outcomes).

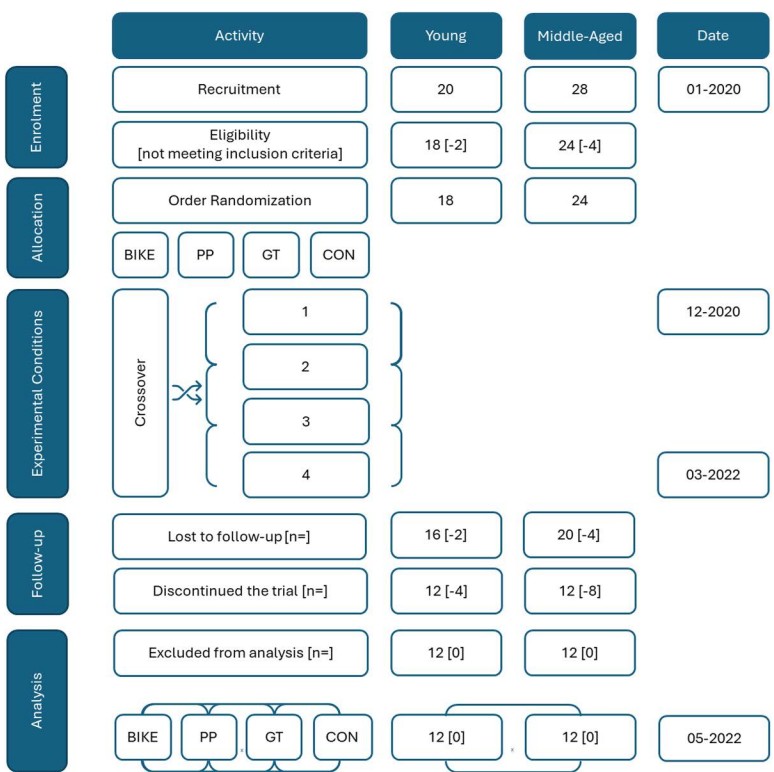

**Fig 1. Flow diagram showing participant flow through each stage of the randomized trial.**

Each session began with 20 minutes of supine rest on a cushioned examination table, with resting energy expenditure (REE) measured by indirect calorimetry (K5, Cosmed, Rome, Italy) and heart rate (HR) and blood pressure (BP) recorded continuously using digital plethysmography (Finapres, NOVA, Finapres Medical Systems, Amsterdam, The Netherlands). This was followed by: 1) Regional assessments of PWV and pulse wave analysis of the carotid, brachial, femoral, and distal arteries on the right side of the body using applanation tonometry (Complior 2.0, Alam Medical; Saint Quentin Fallavier, France); and 2) Assessment of heart rate variability (HRV) and baroreflex sensitivity (BRS) indices using the Finapres NOVA 5 ECG lead module (primary outcomes). Following these assessments, participants engaged in a 45-minute group fitness class, while activity energy expenditure (AEE) was continuously measured by indirect calorimetry (K5, Cosmed, Rome, Italy). In the CON session, participants remained comfortably seated for 45 minutes, maintaining a good posture. After each group fitness class, participants immediately returned to the examination table and recovered in the supine position for 30 minutes, during which local and regional stiffness, HRV, and baroreflex sensitivity (BRS) were re-evaluated at 10-, 20-, and 30-minute intervals during recovery and compared to those at rest. Participants and fitness instructors were blinded to the order of the experimental interventions until arrival at the laboratory. The investigators responsible for outcome measurements were not involved in the delivery of the exercise interventions and remained blinded to the order of the sessions. All sessions were conducted at Ginásio Clube Português during the morning period to minimize potential diurnal variations. Participants were also instructed to maintain their habitual diet, refrain from consuming any food or drink (except water) for 4 hours before the sessions, and avoid alcohol, caffeine, and vigorous exercise for at least 24 hours prior to each session. This trial was retrospectively registered with

ClinicalTrials.gov (ID: NCT06616428). Due to unforeseen disruptions caused by the COVID-19 pandemic and related lockdowns, this registration occurred retrospectively. The pandemic introduced delays in the approval process and recruitment efforts, which hindered prospective registration.

## Group fitness classes

The group fitness classes were characterized by distinct metabolic demands, representing typical classes provided by gyms and health clubs to improve or maintain cardiovascular health. The BIKE group fitness class comprised a rhythmic indoor cycling session, characterized by fluctuations in intensity corresponding to changes in position, music rhythm, cadence, and revolutions per minute. Participants were instructed to strictly follow verbal cues from the instructor, adjusting cycling cadence and resistance as directed. Participants could personalize their intensity by adjusting resistance levels and cycling cadence according to their individual fitness levels, ensuring that both younger and older participants could safely engage in the class at an appropriate effort level. The Pump Power (PP) fitness class entails a comprehensive total-body weight-training program with a focus on improving strength, muscular endurance, and overall fitness of large muscle groups. The class, choreographed to music, involved participants performing a combination of barbell, body-weight exercises, and free-weight plates. Participants selected weights based on the target muscle group for the specific song or track and their individual fitness goals. In this class, participants were encouraged to adjust the weights they used according to their fitness level, ensuring the exercise was both challenging and safe, regardless of the participant's age or prior training experience. The Global Training (GT) class incorporated both aerobic and resistance components, combining athletic movements like running, lunging, and jumping with strength exercises such as barbells, body-weight exercises, and free-weight plates for the large muscle groups. Similar to the other classes, participants were instructed to modify their intensity according to their fitness levels, with the instructor providing guidance on how to adjust movements and intensity to ensure that all participants could perform the exercises safely. Led by the same fitness instructor, and attended by a maximum of 25 other participants, each class lasted 45 minutes and followed a structured format, starting with an opening warm-up, followed by eight tracks targeting specific muscle groups, and concluding with a static stretch cool down (10-12 tracks in total). The choreography of the music played a crucial role as it could influence participant motivation and exercise intensity. To minimize variability, the same music and choreography were standardized across all participants and sessions.

## Measurements

**Body composition.** Height and body weight were measured to the nearest 0.1 cm and 0.1 kg, respectively, using a scale with an attached stadiometer (model 770, Seca; Hamburg, Deutschland). Additionally, body composition was assessed using a seca mBCA 515, which utilizes 4 pairs of electrodes positioned at each hand and foot.

**Cardiorespiratory fitness.** Maximal aerobic capacity was evaluated using a cycle ergometry protocol. Each participant underwent a ramp incremental cycle ergometer test to exhaustion on a calibrated electronically braked cycle ergometer (Monark 839 E, Ergomedic; Monark, Vansbro, Sweden) with a pedal cadence of 70 to 75 rev/min. Initial and incremental workloads were 20-40 W [38,39]. The seat was adjusted to ensure that the participant's legs could reach near full extension during each pedal revolution.

Inspired and expired gases were continuously analyzed with mixing-chamber gas exchange measurements using a portable gas analyzer (K5, Cosmed, Rome, Italy). Before each test, the

oxygen ($O_2$) and carbon dioxide ($CO_2$) analyzers were calibrated using ambient air and standard calibration gases with known concentrations (16.7% $O_2$ and 5.7% $CO_2$). The calibration of the turbine flowmeter of the K5 was performed using a 3-L syringe (Quinton Instruments, Seattle, Wash., USA) following the manufacturer's instructions. Heart rate (HR) was continuously monitored with a chest strap monitor (Garmin, US). Data were assessed in 10-second averages, and peak oxygen uptake ($VO_2$ peak) was defined as the highest 10-second value achieved in the last minute of effort, meeting at least 2 of the following criteria: (1) Attaining ~ 90% of age-predicted maximal HR; (2) Plateau in oxygen uptake ($VO_2$) with an increase in workload (<2.0 mL.kg$^{-1}$.min$^{-1}$); (3) Rating of perceived exertion ≥18 [6–20]; (4) Respiratory exchange ratio (RER) ≥ 1.1; and (5) subjective judgment by the observer that the participant could no longer continue, even after encouragement.

**Energy expenditure.** For the REE assessment, participants were instructed to arrive at the laboratory 60 minutes before each session after a 2-3 hour fast. Following 10 minutes of quiet, seated rest in a dimly lit room, REE was measured for 15 minutes using indirect calorimetry (K5, Cosmed, Rome, Italy) at an environmental temperature and humidity of ±22°C and 40–50%, respectively. During the measurement, outputs of $VO_2$, $CO_2$, respiratory exchange ratio, and ventilation were continuously collected and averaged over 1-minute intervals for subsequent data analysis. The initial 5 minutes of data collection were discarded, and the mean of a 5-minute steady-state interval between the 5th and the 15th minute, where the RER ranged between 0.7 and 1.0, was utilized to calculate REE. Steady state was defined as a 5-minute period with a coefficient of variation for $VO_2$ and $CO_2$ production ≤10% [40]. The mean VO2 and CO2 production of the 5-minute steady states were employed in the Weir equation [41], and the period with the lowest REE was considered.

$VO_2$ was measured throughout the session using the system described above, for total energy expenditure (TEE) measurement. The evaluator marked the exact beginning of each phase of the session to be distinguishable in offline analysis. The K5 from Cosmed (Rome, Italy) is a compact device, easy to attach without constricting the patient's movements. Activity energy expenditure (AEE) was calculated as the difference between TEE and the sum of REE with 0.1 * TEE (assuming the thermic effect of food is ~ 10% of TEE) [42]. These estimates were summed throughout the entire duration of the session. REE was also used to calculate the intensities in metabolic equivalents.

**Regional arterial stiffness.** Arterial stiffness was assessed through PWV in all experimental conditions. Pressure waveforms from the carotid, femoral, radial, and distal arteries were simultaneously captured using applanation tonometry (Complior 2.0, Alam Medical; Saint Quentin Fallavier, France). The distance between pulse sites was directly measured and entered into the Complior Analyse software with the carotid-femoral distance being corrected by a 0.8 factor account for arterial path length differences. Sensors were positioned using specific holders for the carotid and radial arteries, while femoral and distal posterior tibial arteries were held manually. Once 10 carotid pulse waveforms of sufficient quality were obtained, simultaneous recording of carotid and femoral, radial, and distal posterior tibial pressure curves took place for pulse waveforms. The time transit time between two pulse waveforms was then automatically calculated using the foot-to-foot algorithm. The pressure waveform velocity from carotid to femoral (cfPWV) artery, carotid to radial (crPWV) artery, and carotid to distal posterior tibial (cdPWV) artery, were considered as indices of central and peripheral arterial stiffness, respectively.

**Blood pressure.** Resting brachial blood pressures (bSBP and bDBP) were measured in the supine position using an automated oscillometric cuff (HEM-907 XL; Omron Corporation, Japan). Brachial blood pressure was measured twice, and the average of the 2 values was recorded for subsequent analysis. Pulse pressure was calculated as the difference between

systolic and diastolic blood pressure (SBP – DBP). Central blood pressure (cSBP) was assessed by applanation tonometry (Complior 2.0, Alam Medical; Saint Quentin Fallavier, France) in all experimental conditions, derived from right carotid traces obtained during CFPWV assessment. The waveforms were then averaged, and the mean values were extracted from a 15-second window of acquisition. The carotid waveforms were calibrated using mean arterial pressure (MAP) measured immediately before the acquisition.

**Heart rate variability.** Participants were evaluated in a supine position in a quiet climate-controlled room (22-24º C) during all experimental conditions. The R-R intervals were sampled at a frequency of 300 Hz to acquire a digital sequence of R waves using the 5-ECG lead module of the Finapres Nova device (Finapres Medical Systems, Amsterdam, The Netherlands). All data acquisition and off-line analyses were carried out in accordance with the standards set by The Task Force of the European Society of Cardiology and North American Society of Pacing and Electrophysiology [43].

**Heart rate variability data analysis.** All HRV analyses were conducted offline using the FisioSinal software built-in Matlab [44]. After R-R peak detection and semi-automated removal of signal artifacts, 2-minute time series were constructed using cubic spline interpolation, enabling the estimation of both time-domain and spectral power indices [45]. Ectopic heartbeats (mean = 1, standard deviation = 6 beats per minute) were excluded from the final analysis.

The time-domain indices used to characterize HRV were the standard deviation of NN intervals (SDNN) – a measure of overall variability, and the root mean square of the sum of the squares of the differences between NN intervals (RMSSD) – a measure of cardiovagal modulation, both expressed in milliseconds.

Time-frequency domain analysis was performed using the Daubechy-12 discrete wavelet algorithm, allowing the estimation of low-frequency bands (0.04 to 0.15 Hz) and high-frequency bands (0.15 to 0.4 Hz) in absolute and normalized power units. Low-frequency bands reflect both vagal and sympathetic modulation to the heart, whereas the high frequencies reflect only cardiovagal modulation [46]. Wavelet analysis was chosen over fast Fourier transform as it is better suited to characterize the acute responses of the autonomic nervous system during the post-exercise period [47].

**Cardiovagal baroreflex sensitivity.** BRS was estimated using the spontaneous sequence method through the baroreflex module of FisioSinal [44]. In this method, the analysis focused on adjacent oscillations (ramps) in SBP (>1 mmHg) and RR intervals (>4 ms). Beat-to-beat SBP was recorded using finger plethysmography (Finapres® NOVA, Finapres Medical Systems, Amsterdam, The Netherlands).

## Statistical analyses

Based on a medium effect size of 0.154 derived from published changes in aortic PWV within-between modes of exercise [23], an a priori power analysis suggested that 22 participants were required (11 per group) to detect significant differences within-between groups, conditions, and time points (1-β = 80%, α = 0.05). Descriptive statistics for the dependent variables were calculated using R, with the summary tools package. The data are presented as mean (SD) unless stated otherwise.

Parametric inference assumptions, normality and homoscedasticity, were tested with the Shapiro-Wilk and Levene tests, respectively, and by and plot inspection. Independent-sample tests were utilized to compare participant characteristics and experimental conditions. If the normality assumption was not met, we performed non-parametric alternatives, specifically the Mann-Whitney U test, to confirm the robustness of our findings. Changes in dependent variables were examined using mixed linear models fitted with the restricted maximum

likelihood, with Satterthwaite's method used to approximate degrees of freedom for the F-test, employing the R package lmerTest [48]. Fixed effects were defined as time, experimental condition, and group, while a random intercept was defined for each participant.

The R package sjstats [49] was used to calculate partial eta squares ($\eta^2$) for each main effect and interaction, with interpretation based on the benchmarks suggested by Cohen (1988) – small ($\eta^2 < 0.05$), medium ($\eta^2 < 0.25$), and large ($\eta^2 > 0.25$). Covariates, such as HRmax, $\dot{V}O_{2\,peak}$, MAP, inter-beat intervals, sex and fat mass were included separately in the mixed model if necessary. In the presence of significant differences in main effects, post hoc comparisons using Tukey's HSD test were conducted using the R emmeans package [50], and all statistical analyses were carried out using R software, with a significance level ($\alpha$) of 0.05 [51].

## Results

### Characteristics of the participants

The characteristics of the participants are depicted in Table 1. Middle-aged compared to young adults had a higher fat mass (difference (d) = 10.21, 95% CI: 3.63 to 16.78%) and both lower HRmax (d = -33, 95% CI: -43 to -24 b.min$^{-1}$) and $\dot{V}O_{2\,peak}$ (d = - 0.72, 95% CI: -1.41 to -0.02 L.min$^{-1}$).

### Characteristics of the group fitness classes

The physiological characteristics of the group fitness classes are presented in Supplement 1. Exercise-by-group interactions were observed for EE [F (2,43) = 4.48, $p$ = 0.017, $\omega^2$ = 0.13] and the percentage of $\dot{V}O_2$ reserve [F (2,43) = 3.34, $p$ = 0.045, $\omega^2$ = 0.09]. Middle-aged adults attained a lower EE on Bike (d$_{middle-young}$ = -103 kcal, 95% CI: -200 to -6 kcal, $p$ = 0.032) compared to young adults. However, this class in both young and middle-aged adults had a consistently higher EE than PP (d$_{Bike-PP}$ = 129 kcal, 95% CI: 98 to 159 kcal, $p$ < 0.001) and GT (d$_{Bike-GT}$ = 87 kcal, 95% CI: 57 to 117 kcal, $p$ < 0.001). In middle-aged adults, the percentage of $\dot{V}O_2$ reserve attained was different between classes, whereas in young adults PP and GT were similar (Supplement 1). Still, Bike evoked a higher percentage of $\dot{V}O_2$ reserve across groups compared to PP (d$_{Bike-PP}$ = 28%, 95% CI: 23 to 33%, $p$ < 0.001) and GT (d$_{Bike-GT}$ = 18%, 95% CI: 13 to 23%, $p$ < 0.001). Additionally, a main effect of exercise was observed for

Table 1. Characteristics of the participants per age group.

| | Young Adults (n = 12) | Middle-aged Adults (n = 12) | p-value1 |
|---|---|---|---|
| Age (years) | 25 (4) | 60 (7) | <0.01 |
| Height (cm) | 170.3 (8.7) | 167.8 (9.3) | 0.50 |
| Weight (kg) | 67.2 (11.3) | 71.7 (18.5) | 0.48 |
| Body mass index (kg.m$^{-2}$) | 23.0 (1.9) | 25.2 (4.5) | 0.14 |
| Waist circumference (m) | 0.80 (0.09) | 0.91 (0.11) | 0.02 |
| Fat mass (%) | 19.5 (6.4) | 29.7 (8.9) | 0.04 |
| Fat mass (kg) | 12.8 (4.1) | 21.5 (8.8) | 0.07 |
| HR$_{max}$ (b.min$^{-1}$) | 184 (4) | 150 (15) | <0.01 |
| HRR (b.min$^{-1}$) | 62 (13) | 53 (14) | 0.20 |
| VO$_2$ peak (mL.kg$^{-1}$.min$^{-1}$) | 40.9 (8.6) | 27.9 (6.6) | <0.01 |
| VO$_{2\,peak}$ (L.min$^{-1}$) | 2.06 (0.80) | 2.78 (0.85) | 0.04 |

Data depicted as mean (SD). Abbreviations: HR$_{max}$, maximal heart rate: HRR, heart rate recovery. 1, Welch's t test was used to test group differences.

HR reserve % suggesting that both groups worked at higher HR reserve during Bike when compared to PP ($d_{Bike-PP}$ = 13%, 95% CI: 5 to 20%, $p$ = 0.011) and GT ($d_{Bike-GT}$ = 10%, 95% CI: 2 to 17%, $p$ < 0.001). No harms or adverse events were reported or verified during the group fitness classes.

### Resting blood pressure, arterial stiffness and cardiovagal modulation

Middle-aged adults exhibited a higher cSBP (d = 12 mmHg, 95% CI: 1 to 22 mmHg), bSBP (d = 16 mmHg, 95% CI: 7 to 26 mmHg) and bDBP (d = 14 mmHg, 95% CI: 10 to 19 mmHg) than young adults at rest. Central and peripheral PWVs were not different between groups (Table 2). Middle-aged showed a lower Ln-SDNN (d = -0.42 ms, 95% CI: -0.62 to -0.22 ms), Ln-RMSSD (d = -0.52 ms, 95% CI: -0.87 to -0.17 ms) and Ln-HF (d = -1.14 ms², 95% CI -1.88 to -0.39 ms²) but not BRS (d = -0.55 ms/mmHg, 95% CI: -2.41 to 1.32 ms/mmHg), compared to young adults. Controlling for sex, fat mass and $VO_{2\,peak}$ did not abolish these group effects.

### Blood pressure and arterial stiffness after the group fitness classes

Group-by-time interactions were observed in cSBP [F (3, 326) = 3.80, $p$ = 0.011, $\omega^2$ = 0.02] and bSBP (Fig 2) following the group fitness classes. These indicated that only middle-aged adults showed reductions 20-min into recovery in both cSBP ($d_{post20-bas}$ = -8 mmHg, 95% CI: -14 to -2 mmHg, $p$ = 0.002) and bSBP ($d_{post20-bas}$ = -7 mmHg, 95% CI: -12 to -2 mmHg, $p$ = 0.004). Exercise-by-time interactions were also observed in bDBP (Table 2). Middle-aged adults exhibited similar reductions in bDBP at the 10-min time point following Bike ($d_{post20-bas}$ = -6 mmHg, 95% CI: -11 to -1 mmHg, $p$ = 0.027) and PP ($d_{post10-bas}$ = -7 mmHg, 95% CI: -11 to -1 mmHg, $p$ = 0.048) but not GT ($d_{post10-bas}$ = -3 mmHg, 95% CI: -3 to -8 mmHg, $p$ = 0.543), compared to young adults. Controlling for between-group differences in $VO_{2\,peak}$, fat mass and biological sex did not change the above results. cfPWV, crPWV and cdPWV remained unchanged after fitness classes (Table 3), even after adjustment for changes in MAP.

### Cardiovagal modulation after the group fitness classes

Exercise-by-time interactions were observed for Ln-SDNN (Table 3), Ln-RMSSD (Fig 3), pNN50 (Table 3), Ln-HF (Fig 3), Ln-LF (Table 3), LF/HF (Table 3) and BRS (Table 2) following the group fitness classes. These suggested similar reductions in Ln-SDNN ($d_{post10-bas}$ = -0.51 ms, 95% CI: -0.64 to -0.37 ms, $p$ < 0.001), Ln-RMSSD ($d_{post10-bas}$ = -0.90 ms, 95% CI: -1.09 to -0.72 ms, $p$ < 0.001), and BRS ($d_{post30-bas}$ = -4.03 ms/mmHg, 95% CI: -5.36 to -2.71 ms/mmHg, $p$ < 0.001), and increases in LF/HF ($d_{post10-bas}$ = 1.44, 95% CI: 0.50 to 2.39, $p$ < 0.001) immediately after all fitness classes, which persisted 30-min into recovery only after Bike (Ln-SDNN: $d_{post30-bas}$ = -0.48 ms, 95% CI: -0.71 to -0.25 ms, $p$ < 0.001; Ln-RMSSD: $d_{post30-bas}$ = -0.57 ms, 95% CI: -0.89 to -0.26 ms, $p$ < 0.001). All fitness classes similarly reduced pNN50 ($d_{post30-bas}$ = -16%, 95% CI: -21 to -12%, $p$ < 0.001), Ln-HF ($d_{post30-bas}$ = -1.03 ms², 95% CI: -1.40 to -0.67 ms², $p$ < 0.001), and Ln-LF ($d_{post30-bas}$ = -0.39 ms², 95% CI: -0.71 to -0.07 ms², $p$ < 0.001) during early recovery in both young and middle-aged adults. Controlling for inter-beat interval, $VO_{2\,peak}$, fat mass, and sex did not change the above results. Similar results were noted with untransformed HRV data (Supplement 2).

## Discussion

The present study investigated the acute effects of three commercially available group exercise classes on arterial stiffness and cardiovagal modulation in both young and middle-aged adults. The main findings were that only middle-aged adults showed reductions in central and

**Table 2. Blood pressure and arterial stiffness responses after group fitness classes in young and middle-aged adults.**

| | Young Adults | | | | Middle-aged Adults | | | | Time | Ex | Gr | Ex*Time | Ex*Time*Gr |
|---|---|---|---|---|---|---|---|---|---|---|---|---|---|
| | Rest | 10 | 20 | 30 | Rest | 10 | 20 | 30 | p (w2) | p (w2) | p (w2) | p (w2) | p (w2) |
| **bDBP, mmHg** | | | | | | | | | 0.011 (0.02) | <0.001 (0.05) | <0.001 (0.66) | 0.041 (0.03) | 0.19 (0.00) |
| CON | 64 (6) | 62 (4) | 67 (5) | 66 (5) | 81 (10) | 83 (10) | 83 (9) | 81 (9) | | | | | |
| BIKE | 67 (8) | 64 (6)* | 64 (7) | 63 (8) | 81 (10) | 71 (9)* | 74 (12) | 77 (10) | | | | | |
| PP | 65 (8) | 61 (8)* | 63 (6)* | 60 (6)* | 82 (19) | 75 (9)* | 71 (5)* | 73 (7)* | | | | | |
| GT | 64 (8) | 63 (6) | 63 (6) | 64 (8) | 78 (14) | 74 (8) | 80 (9) | 81 (6) | | | | | |
| **bMAP, mmHg** | | | | | | | | | 0.003 (0.03) | 0.002 (0.04) | <0.001 (0.59) | 0.057 (0.02) | 0.18 (0.01) |
| CON | 80 (6) | 78 (4) | 83 (3) | 81 (5) | 98 (12) | 99 (13) | 99 (11) | 99 (11) | | | | | |
| BIKE | 82 (7) | 80 (5) | 79 (7) | 78 (5) | 99 (11) | 87 (10) | 90 (13) | 94 (12) | | | | | |
| PP | 80 (6) | 79 (7) | 79 (5) | 76 (5) | 99 (20) | 91 (10) | 89 (7) | 92 (10) | | | | | |
| GT | 81 (7) | 80 (6) | 79 (7) | 79 (7) | 95 (15) | 88 (10) | 96 (12) | 98 (8) | | | | | |
| **cfPWV, m/s** | | | | | | | | | 0.037 (0.02) | 0.10 (0.00) | 0.12 (0.06) | 0.67 (0.00) | 0.94 (0.00) |
| CON | 6.75 (0.95) | 6.91 (1.13) | 6.68 (0.66) | 6.69 (0.53) | 7.58 (1.88) | 7.38 (2.27) | 7.19 (1.32) | 7.42 (1.54) | | | | | |
| BIKE | 6.42 (0.93) | 6.65 (0.74) | 6.52 (0.69) | 6.33 (0.50) | 6.91 (1.68) | 7.67 (1.29) | 7.33 (1.61) | 7.24 (1.07) | | | | | |
| PP | 6.75 (0.61) | 6.85 (0.91) | 6.89 (0.88) | 6.71 (1.25) | 7.34 (1.98) | 7.81 (1.60) | 7.55 (1.35) | 7.53 (1.93) | | | | | |
| GT | 6.58 (0.90) | 6.90 (0.93) | 6.66 (0.70) | 6.68 (0.66) | 6.78 (1.41) | 7.68 (1.72) | 7.24 (1.67) | 7.65 (2.24) | | | | | |
| **crPWV, m/s** | | | | | | | | | 0.79 (0.00) | 0.65 (0.00) | 0.054 (0.12) | 0.93 (0.00) | 0.99 (0.00) |
| CON | 8.21 (1.60) | 8.04 (0.45) | 7.95 (1.54) | 8.30 (1.47) | 9.11 (1.54) | 9.46 (2.27) | 9.71 (2.13) | 9.42 (2.02) | | | | | |
| BIKE | 8.63 (1.20) | 8.22 (0.95) | 7.81 (1.27) | 8.26 (0.84) | 8.90 (1.93) | 9.34 (1.96) | 9.39 (1.60) | 9.23 (1.46) | | | | | |
| PP | 8.70 (1.32) | 8.29 (1.07) | 8.28 (1.37) | 8.28 (1.29) | 8.51 (2.61) | 9.44 (2.37) | 9.19 (2.17) | 9.12 (1.87) | | | | | |
| GT | 8.70 (1.52) | 8.72 (1.24) | 8.07 (1.39) | 8.53 (1.32) | 8.63 (1.39) | 9.63 (2.44) | 9.47 (1.86) | 9.53 (1.98) | | | | | |
| **cdPWV, m/s** | | | | | | | | | 0.36 (0.00) | 0.26 (0.00) | 0.054 (0.12) | 0.74 (0.00) | 0.83 (0.00) |
| CON | 7.92 (0.64) | 8.04 (0.69) | 8.18 (0.69) | 8.06 (0.62) | 8.78 (1.25) | 8.78 (1.57) | 8.82 (1.15) | 8.82 (0.98) | | | | | |
| BIKE | 8.04 (0.83) | 7.87 (0.58) | 7.71 (0.54) | 7.82 (0.79) | 8.89 (1.57) | 8.62 (1.56) | 8.37 (1.16) | 8.53 (1.54) | | | | | |
| PP | 8.26 (0.44) | 7.77 (0.80) | 7.98 (0.65) | 7.89 (0.88) | 8.86 (1.42) | 8.94 (1.81) | 8.83 (1.28) | 8.55 (1.33) | | | | | |
| GT | 8.27 (0.80) | 7.92 (0.72) | 7.72 (0.71) | 7.92 (0.59) | 8.51 (1.30) | 8.78 (1.58) | 8.65 (1.70) | 8.71 (1.71) | | | | | |
| **BRS, ms/ mmHg** | | | | | | | | | <0.001 (0.16) | <0.001 (0.12) | 0.473 (0.00) | 0.005 (0.04) | 0.056 (0.00) |
| CON | 11.82 (4.07) | 9.57 (5.02) | 9.61 (5.72) | 11.12 (4.00) | 9.27 (4.62) | 11.05 (3.99) | 10.64 (3.41) | 10.84 (4.09) | | | | | |
| BIKE | 10.36 (2.20) | 6.16 (1.51)* | 6.27 (1.55)* | 5.85 (1.80)* | 8.67 (4.22) | 4.31 (1.72)* | 5.00 (2.18)* | 5.41 (2.17)* | | | | | |
| PP | 9.21 (2.91) | 7.10 (3.54)* | 7.11 (2.29)* | 7.38 (2.95)* | 15.20 (7.71) | 4.89 (2.41)* | 5.57 (2.13)* | 6.32 (3.14)* | | | | | |
| GT | 13.67 (7.40) | 7.17 (4.20)* | 6.27 (2.46)* | 6.77 (3.42)* | 11.37 (4.23) | 5.43 (2.48)* | 5.75 (2.40)* | 5.78 (2.50)* | | | | | |

Data presented as mean (SD). Abbreviations, bDBP, brachial diastolic blood pressure; bMAP, brachial mean arterial pressure; cSBP, central systolic blood pressure; cfPWV, carotid-femoral pulse wave velocity; crPWV, carotid-radial pulse wave velocity; cdPWV, carotid-dorsalis pedis pulse wave velocity.

* Different from resting time point (p < 0.05).

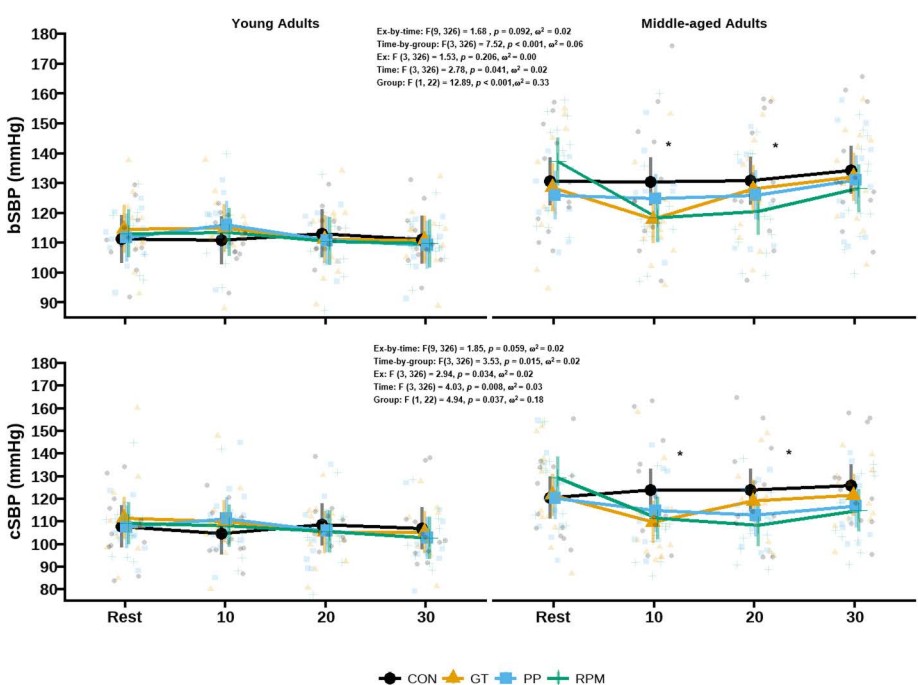

**Fig 2. Central and peripheral systolic blood pressure responses after the group fitness classes in young and middle-aged adults.** Abbreviations: bDBP, brachial diastolic blood pressure; cSBP, central systolic blood pressure.

brachial blood pressure 20-min into recovery following the group fitness classes, but there were no changes in arterial stiffness. All fitness classes similarly reduced time and frequency domain metrics of HRV both in young and middle-aged adults during early recovery, but reductions persisted 30-min into recovery only after Bike. Thus, under an ecologically valid exercise setting, and in line with our hypothesis, the observed acute effects varied based on age, measurement time point, and, in the case of autonomic recovery, exercise mode.

While the impact of exercise on arterial stiffness has been extensively demonstrated [29,52], the immediate effects of different exercise modalities on arterial stiffness remain uncertain. A meta-analysis from 2018 indicated that acute aerobic exercise did not alter cfPWV in healthy adults, while resistance exercise increased it [21]. However, a review by Saz Lara in 2021 [20] found that aerobic exercise effectively reduced cfPWV between 30 minutes to 24 hours post-exercise, with no significant changes observed after resistance exercise. In addition, peripheral PWV showed a significant decrease after exercise until 60 min into recovery. Notably, extensive heterogeneity is observed within both meta-analyses, leading to conflicting results across numerous studies concerning exercise type and the duration of its acute effects on central and peripheral arterial stiffness. In employing an ecological model, the current study determined that neither central nor peripheral arterial stiffness changed following the group fitness classes in young and middle-aged adults. This inconsistency among studies arises from the transient nature of exercise effects, the arterial segment evaluated [53], the intensity [20], the measurement time point [53], the age of the participants [54,55] and their fitness level [56]. Although still not well defined, post-exercise central PWV is elevated in those with lower as compared to those with higher cardiorespiratory fitness [56]. In the present study, only 4 participants in the young group were classified as unfit, while the prevalence was higher in the middle-aged group (n = 7). The generally high fitness levels of the participants may explain the lack of significant arterial responses. Additionally, both young

**Table 3. Cardiovagal modulation response after the group fitness classes in young and middle-aged adults.**

| | Young Adults | | | | Middle-aged Adults | | | | Time | Ex | Gr | Ex*Time | Ex*Time*Gr |
|---|---|---|---|---|---|---|---|---|---|---|---|---|---|
| | Rest | 10 | 20 | 30 | Rest | 10 | 20 | 30 | p (w2) CON | p (w2) | p (w2) | p(w2) | p (w2) |
| **Ln-IBI, ms** | | | | | | | | | **<0.001 (0.49)** | **<0.001 (0.60)** | 0.20 (0.03) | **<0.001 (0.28)** | 0.67 (0.00) |
| CON | 6.89 (0.13) | 6.89 (0.12) | 6.89 (0.12) | 6.89 (0.11) | 6.96 (0.13) | 7.00 (0.12) | 7.00 (0.13) | 7.02 (0.13) | | | | | |
| BIKE | 6.83 (0.14) | 6.60 (0.12)* | 6.63 (0.10)* | 6.66 (0.11)* | 6.91 (0.13) | 6.65 (0.15)* | 6.70 (0.15)* | 6.75 (0.14)* | | | | | |
| PP | 6.91 (0.15) | 6.64 (0.16)* | 6.69 (0.17)* | 6.72 (0.14)* | 6.90 (0.14) | 6.69 (0.15)* | 6.71 (0.14)* | 6.75 (0.11)* | | | | | |
| GT | 6.87 (0.15) | 6.66 (0.19)* | 6.66 (0.17)* | 6.71 (0.18)* | 6.95 (0.12) | 6.68 (0.17)* | 6.73 (0.18)* | 6.77 (0.16)* | | | | | |
| **Ln-SDNN, ms** | | | | | | | | | **<0.001 (0.15)** | **<0.001 (0.19)** | **<0.001 (0.43)** | **<0.001 (0.17)** | 0.96 (0.00) |
| CON | 3.94 (0.30) | 4.25 (0.19) | 4.21 (0.23) | 4.09 (0.29) | 3.60 (0.35) | 3.85 (0.27) | 3.68 (0.34) | 3.78 (0.35) | | | | | |
| BIKE | 4.15 (0.31) | 3.52 (0.21)* | 3.71 (0.31)* | 3.73 (0.31)* | 3.83 (0.30) | 3.16 (0.36)* | 3.17 (0.48)* | 3.29 (0.45)* | | | | | |
| PP | 4.10 (0.21) | 3.61 (0.50)* | 3.80 (0.57)* | 3.86 (0.45) | 3.58 (0.42) | 3.21 (0.45)* | 3.31 (0.26)* | 3.51 (0.53) | | | | | |
| GT | 4.07 (0.25) | 3.72 (0.35)* | 3.71 (0.48)* | 3.87 (0.40) | 3.72 (0.43) | 3.21 (0.36)* | 3.09 (0.45)* | 3.53 (0.61) | | | | | |
| **pNN50, %** | | | | | | | | | **<0.001 (0.24)** | **<0.001 (0.26)** | **<0.001 (0.42)** | **<0.001 (0.19)** | 0.25 (0.00) |
| CON | 29 (15) | 38 (18) | 34 (14) | 34 (21) | 9 (8) | 16 (10) | 16 (13) | 16 (12) | | | | | |
| BIKE | 33 (16) | 5 (6)* | 8 (7)* | 11 (10)* | 10 (12) | 1 (2)* | 2 (4)* | 2 (3)* | | | | | |
| PP | 38 (19) | 10 (17)* | 17 (18)* | 21 (19)* | 16 (21) | 2 (3)* | 1 (3)* | 2 (4)* | | | | | |
| GT | 42 (20) | 14 (14)* | 12 (15)* | 15 (15)* | 16 (15) | 1 (3)* | 2 (5)* | 4 (6)* | | | | | |
| **Ln-LF, ms²** | | | | | | | | | **<0.001 (0.08)** | **<0.001 (0.11)** | **<0.001 (0.46)** | **<0.001 (0.08)** | 0.85 (0.00) |
| CON | 6.76 (0.87) | 7.29 (0.63) | 7.18 (0.65) | 6.82 (0.73) | 6.07 (1.10) | 6.33 (0.67) | 5.98 (0.89) | 6.16 (0.82) | | | | | |
| BIKE | 7.27 (0.93) | 6.19 (0.61)* | 6.74 (0.66)* | 6.68 (0.62)* | 6.15 (0.93) | 4.78 (0.84)* | 5.05 (1.23)* | 5.37 (1.02)* | | | | | |
| PP | 6.83 (0.89) | 6.13 (0.90)* | 6.49 (1.19)* | 6.68 (0.88)* | 5.68 (0.69) | 5.13 (0.92)* | 5.11 (0.58)* | 5.41 (1.07)* | | | | | |
| GT | 6.72 (0.96) | 6.52 (1.08)* | 6.26 (0.98)* | 6.55 (0.89)* | 6.23 (0.85) | 5.10 (1.02)* | 4.97 (0.89)* | 5.91 (1.50)* | | | | | |
| **Ln-LF/HF** | | | | | | | | | **<0.001 (0.12)** | **<0.001 (0.11)** | 0.59 (0.00) | **0.004 (0.050)** | 0.77 (0.00) |
| CON | 1.03 (0.13) | 1.04 (0.08) | 1.04 (0.09) | 1.01 (0.13) | 1.11 (0.28) | 1.05 (0.15) | 1.06 (0.21) | 1.07 (0.18) | | | | | |
| BIKE | 1.10 (0.21) | 1.27 (0.22)* | 1.24 (0.20)* | 1.20 (0.20)* | 1.10 (0.23) | 1.43 (0.41)* | 1.30 (0.39)* | 1.18 (0.20)* | | | | | |
| PP | 1.01 (0.19) | 1.36 (0.62)* | 1.25 (0.47)* | 1.16 (0.31)* | 1.06 (0.34) | 1.30 (0.43)* | 1.18 (0.28)* | 1.25 (0.38)* | | | | | |
| GT | 0.98 (0.19) | 1.24 (0.40)* | 1.29 (0.53)* | 1.20 (0.31)* | 1.04 (0.11) | 1.44 (0.49)* | 1.37 (0.44)* | 1.28 (0.44)* | | | | | |

Data depicted as mean (SD). Abbreviations, CON, control; PP, pump power; GT, global training; IBI, interbeat interval; SDNN, standard deviation of normal-to-normal RR intervals. Abbreviations, IBI, interbeat interval; SDNN, standard deviation of normal-to-normal NN intervals, pNN50, percentage of RR intervals differing > 50 ms; LF, low-frequency power band; HF, high frequency power band.

* Different from resting time point ($p < 0.05$).

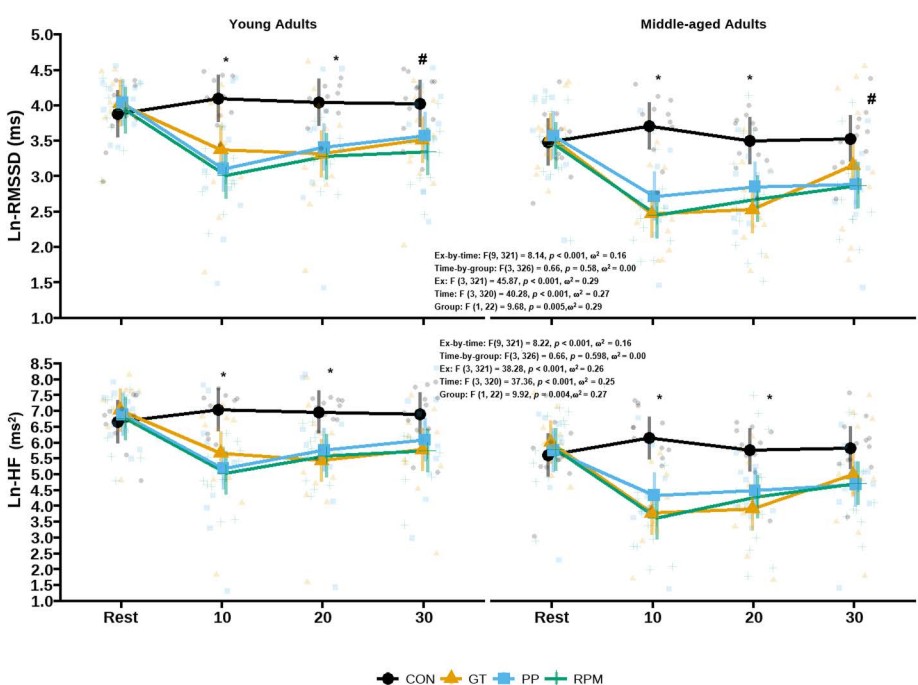

**Fig 3. Selected time and frequency domain indices of heart rate variability response after the group fitness classes in young and middle-aged adults.**

and middle-aged adults had normal central PWV at rest, suggesting a potential ceiling effect, which may limit further reductions following exercise. Finally, the reviewed findings primarily originate from studies focused on exercises conducted in laboratory settings rather than activities that are presented in the format of group fitness classes to the general public. Group fitness classes include rest breaks between sets, a feature absent in most laboratory exercise models and incorporate 5-minute static stretches at the end, potentially impacting post-exercise stiffness results.

Still, a window of benefit [26] in central and peripheral blood pressure was observed in middle-aged adults regardless of the mode of exercise. There is extensive evidence that a single bout of aerobic or resistance exercise performed alone or combined can reduce central [21,57] and peripheral [58,59] blood pressure compared with control values in young and middle-aged adults. This phenomenon is referred to as postexercise hypotension (PEH) and typically reaches its maximum magnitude around 30-min after exercise in people with normal blood pressure or hypertension, which aligns with the intensity-dependent biphasic response of the autonomic nervous system [28]. The clinical importance of PEH phenomenon has been acknowledged [60] and is clinically important because: (a) it occurs immediately; (b) a person does not have to be physically fit to experience its blood pressure control benefits; (c) PEH is strongly correlated with the magnitude of the blood pressure reduction that results from long-term exercise training, suggesting that the BP benefits that result from exercise training may be due to PEH [58]. In the present study, these transient changes in blood pressure after acute exercise were shown independent of the lack of changes in PWV and concomitant to reduced parasympathetic modulation [58,61].

Several studies conducted on healthy people have investigated the isolated effects of aerobic exercise [30,62,63] and resistance exercise [64–66] on cardiac autonomic control. Niemela et al. [67], for example, investigated the association between the exercise mode and recovery pattern of

HRV after aerobic exercise (50% of maximal power output on a cycle ergometer) and resistance exercise (light intensity: 30% of 1RM and heavy intensity: 80% of 1RM) in 12 healthy men. The authors concluded that the vagally mediated HF power was reduced and LF/HF ratio augmented 30 min after heavy resistance exercise compared with control measurements. Heffernan et al. [68] investigated the acute responses of cardiac autonomic control assessed by HRV after aerobic exercise (30 min of upright stationary cycling at 65% $Vo_2$peak) and resistance exercise (3 x 10RM x 8 exercises) in 14 healthy and moderately active men. Autonomic cardiovascular control of the heart was not fully regained after 30 min of either acute aerobic or resistance exercise. There were similar reductions in normalized HF power and similar increases in normalized LF power after both bouts of exercise. The LF/HF ratio increased similarly after both bouts, suggesting a comparable shift towards a state of sympathetic predominance. However, greater reductions in total power were noted after acute resistance exercise, suggesting greater reductions in parasympathetic modulation. In this study, we observed increased sympathetic and reduced parasympathetic activity during recovery, thus our findings using an ecological model were somewhat consistent with prior results. However, unlike the findings from Niemela et al. [67] and Heffernan et al [68], indices of HRV were fully restored 30 minutes following resistance or combined exercise, except for Bike. There are mechanisms involved in post-exercise autonomic regulation, such as central command and cardiopulmonary and arterial baroreflex activity, which are thought to be influenced by exercise intensity in middle-aged [62] and older adults [69]. Thus, slower vagal reactivation was expected during the recovery period following BIKE, because the average %HRR and energy expenditure was higher compared to the other group fitness classes. Overall, our findings are supportive that an increased heart rate, rather than the amount of oxidative or non-oxidative energy contribution, or increased EE, are key factors supporting lower vagal HRV measurements during post-exercise recovery.

Our data also reinforce previous premises that acute resistance exercises, performed at moderate-to-high intensity, such as those used in PP, influence post-exercise autonomic control, increases sympathetic dominance with a concomitant attenuation of BRS [25,67]. While others have failed to find these changes in standardized resistance exercise settings, our findings suggest that acute resistance exercise can elicit similar cardiovagal modulation and delayed BRS recovery pattern as aerobic exercise, at least in apparently healthy participants within an ecologically valid model of exercise. On the other hand, previous studies also highlight that total exercise volume is a major determinant of post-exercise changes in autonomic control [64,70]. However, although energy expenditure was higher in the Bike group fitness class, no differences in cardiovagal modulation were observed between exercise modalities.

## Practical implications

Our findings introduce several important distinctions to the existing literature: 1) Previous research primarily focused on the immediate acute effects in younger participants, largely excluding middle-aged adults; 2) Exercise duration in prior studies ranged from 10 to 60 minutes, whereas our classes were consistently 45 minutes; 3) Group fitness classes are intermittent, with rest intervals between music tracks; and 4) The amount of exercised muscle mass is a key component of exercise volume. Group fitness classes tend to engage the whole body, rather than isolating muscle groups.

The practical implications of our study underscore the safety and efficacy of group fitness classes for cardiovascular health, particularly in ecologically valid settings. Importantly, our findings suggest that acute cardiovascular responses to different group exercise modalities—including indoor cycling, resistance training, and combined exercise training - do not pose risks to participants. Specifically, arterial stiffness remained unchanged after all group fitness classes, indicating that these activities do not adversely affect arterial health.

Furthermore, reductions in blood pressure and the recovery of heart rate variability within 30 minutes highlight potential cardiovascular benefits, especially for middle-aged adults. However, it's crucial to note that recovery of autonomic function differed across exercise modes, with the indoor cycling class (Bike) showing prolonged effects on heart rate variability compared to resistance training and combined exercise modalities. This suggests that exercise mode can influence the duration of autonomic recovery, a key consideration when designing exercise programs for cardiovascular health. The phenomenon of postexercise hypotension, observed particularly in middle-aged adults, not only reflects acute cardiovascular responses but also hints at potential long-term benefits for blood pressure management and cardiovascular risk reduction.

## Limitations

This study is not devoid of limitations. Monitoring post-intervention responses only every 10 minutes for the initial 30 minutes may have precluded the detection of subtle changes or those occurring beyond this timeframe (e.g., >30 min – 72 h). Additionally, due to the constraints of measurement timings, immediate measurements, e.g., 0–5 minutes following exercise, were not obtained. Given that all participants were physically active, the outcomes may deviate from those observed in physically inactive and/or clinical populations, necessitating further investigation. The influence of the menstrual cycle and menopause variability in young and middle-aged women was not controlled for, which could potentially impact arterial measurements [71], but this contention has been refuted [72]. Furthermore, respiratory sinus arrhythmia, which could have interfered with HRV and BRS, was not controlled for. Despite recommendations to employ rhythmic breathing to mitigate the influence of respiration on autonomic parameters [46], the literature suggests no significant effect on post-exercise HRV [73]. Additionally, our study lacked invasive measurements such as blood lactate and plasma catecholamine concentrations, which could furnish further insight into sympathetic activity and skeletal muscle recovery following exercise. Moreover, it is noteworthy to mention that indirect calorimetry, while accurate for assessing metabolic intensities, economy, efficiency, and energy expenditure during steady-state exercise [74], was employed during group fitness classes, potentially impacting its precision in this setting.

## Supplement Information:

**S1 Supplement. Physiological demands of the group fitness classes** . Data depicted as mean (SD). Abbreviations: PP, Pump Power; GT; Global Training; REE, resting energy expenditure; EE, energy expenditure; TEE; total energy expenditure; HR, heart rate; HRR, heart rate reserve; VO2 RR, oxygen uptake reserve; MET, metabolic equivalent. #Different from Young Adults in the corresponding fitness class ($p < 0.05$); * Different from BIKE within groups ($p < 0.05$). †Different from PP within groups ($p < 0.001$).
(PDF)

**S2 Supplement. Untransformed cardiovagal modulation response data after the group fitness classes.** Data depicted as mean (SD). Abbreviations, CON, control; PP, pump power; GT, global training; IBI, interbeat interval; SDNN, standard deviation of normal-to-normal RR intervals. Abbreviations, IBI, interbeat interval; SDNN, standard deviation of normal-to-normal NN intervals, LF, low-frequency power band; HF, high frequency power band. * Different from resting time point ($p < 0.05$).
(XLSX)

## Acknowledgements

The authors express their sincere gratitude and extend acknowledgements to all study participants and their families for their invaluable time and energy dedicated to this research endeavor. Furthermore, appreciation is extended to João Pejapes and Alexandra Silva from the Exercise and Health Department of Ginásio Clube Português for their substantial logistical support and personal contributions, which significantly contributed to the successful completion of this study.

## Author contributions

**Conceptualization:** Xavier Melo, João L. Marôco, Bo Fernhall, Helena Santa-Clara.

**Data curation:** Adma Lopes, Raquel Coelho, Bruno Simão, Isabel Oliveira.

**Formal analysis:** Adma Lopes, Raquel Coelho, João L. Marôco.

**Investigation:** Xavier Melo, Adma Lopes, Raquel Coelho, Bruno Simão, Isabel Oliveira.

**Methodology:** Xavier Melo.

**Project administration:** Xavier Melo.

**Software:** Sérgio Laranjo.

**Supervision:** Xavier Melo.

**Validation:** Sérgio Laranjo, Bo Fernhall, Helena Santa-Clara.

**Visualization:** João L. Marôco.

**Writing – original draft:** Xavier Melo.

**Writing – review & editing:** João L. Marôco, Sérgio Laranjo, Bo Fernhall, Helena Santa-Clara.

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
