## [Decision Letter · Decision Letter 0]

13 Dec 2024

PONE-D-24-44766Acute effects of Commercial Group Exercise Classes on Arterial Stiffness and Cardiovagal Modulation in Healthy Young and Middle-aged Adults: a crossover randomized trialPLOS ONE

Dear Dr. Melo,

Thank you for submitting your manuscript to PLOS ONE. After careful consideration, we feel that it has merit but does not fully meet PLOS ONE’s publication criteria as it currently stands. Therefore, we invite you to submit a revised version of the manuscript that addresses the points raised during the review process.

**ACADEMIC EDITOR:**

We look forward to receiving your revised manuscript.

Kind regards,

Hidetaka Hamasaki

Academic Editor

PLOS ONE

Journal Requirements:

2. Please include captions for your Supporting Information files at the end of your manuscript, and update any in-text citations to match accordingly. Please see our Supporting Information guidelines for more information: http://journals.plos.org/plosone/s/supporting-information .

Reviewers' comments:

Reviewer's Responses to Questions

**Comments to the Author**

1. Is the manuscript technically sound, and do the data support the conclusions?

Reviewer #1: Yes

Reviewer #2: Yes

2. Has the statistical analysis been performed appropriately and rigorously? 

Reviewer #1: Yes

Reviewer #2: Yes

3. Have the authors made all data underlying the findings in their manuscript fully available?

Reviewer #1: Yes

Reviewer #2: Yes

4. Is the manuscript presented in an intelligible fashion and written in standard English?

Reviewer #1: Yes

Reviewer #2: Yes

5. Review Comments to the Author

Reviewer #1: The authors recruited 12 young and 12 middle-aged adults to investigate the acute effects of commercially available group fitness classes on arterial stiffness and vagal-related heart rate variability (HRV). The results showed the consistent overall cardiovascular effects but different automatic recovery between exercise modes, suggesting considering age, exercise modality, and recovery time for exercise prescription.

1. Abstract. please spell out for the first time use of any abbreviations, e.g., BRS

2. Line 322. Normality was tested and independent t-sample tests were utilized. What if normality assumption was not met?

3. Line 331. “Covariates were included in the mixed model if necessary”. Please clarify what covariates were included in the analysis.

4. Line 338. Fat mass, HRmax and VO2 peak were found differently between two groups. In the following analysis, it seems not all three being adjusted in the analysis. What if these were adjusted in the analysis? Were the results and conclusions retained?

Reviewer #2: In the present study, the authors investigated the acute effects of 3 group exercise classes on arterial stiffness and cardiovascular modulation in both young and middle-aged adults. The manuscript is well-written and organized. However, there are some concerns regarding the transparency of the interventions which must be addressed by the authors. I have provided some comments for authors’ consideration.

Page 9, Line 166; Why the body composition and cardiorespiratory fitness were just measured in the CON session?

Page 9, Line 183; what was the rationale of blinding the participants and fitness instructor until arrival at the lab? Did the individual measuring the outcome measures was blinded regarding the order of exercise intervention?

Did you have any control over the diet 24 or 48 hours before performing each experimental session?

Page 10, Line 193; There are some concerns about the details of these 3 interventions. There is missing information on how each training session was personalized for each individual. Typically, the participants have different fitness level and accordingly, each training session must be personalized for them based on their fitness level. How did the author control this in each session. It’s crucial to provide detailed information in this regard. For instance, did a participant aged more than 48 years old perform the same exercise as a 21 years old participant, without considering their age and fitness level?

Page 11, Line 210; What if the music performed in each session had a confounding effect on participants? The same music could have different effect on each individual regarding the emotions and motivations.

Page 11, Line 222; What was the duration of each stage in the incremental cycling test? Please provide a reference for this test protocol.

6. PLOS authors have the option to publish the peer review history of their article (what does this mean? ). If published, this will include your full peer review and any attached files.

**Do you want your identity to be public for this peer review?** For information about this choice, including consent withdrawal, please see our Privacy Policy .

Reviewer #1: No

Reviewer #2: No

---

## [Author Response · Author response to Decision Letter 1]

26 Dec 2024

Review Report

Thank you for the opportunity to revise our manuscript entitled “Acute effects of Commercial Group Exercise Classes on Arterial Stiffness and Cardiovagal Modulation in Healthy Young and Middle-aged Adults: a crossover randomized trial.” We have responded below to each comment from the reviewers and believe we have now submitted an improved version of the manuscript. Significant changes include clarifications on statistical methods and methodological procedures, such as blinding, fitness classes, and exercise testing. Specific changes incorporated in the revised manuscript are highlighted in red. Please see our responses to your suggestions below.

Reviewer #1

The authors recruited 12 young and 12 middle-aged adults to investigate the acute effects of commercially available group fitness classes on arterial stiffness and vagal-related heart rate variability (HRV). The results showed the consistent overall cardiovascular effects but different automatic recovery between exercise modes, suggesting considering age, exercise modality, and recovery time for exercise prescription.

1. Abstract. please spell out for the first time use of any abbreviations, e.g., BRS

Addressed as suggested.

2. Line 322. Normality was tested and independent t-sample tests were utilized. What if normality assumption was not met?

If the normality assumption was not met, we performed non-parametric alternatives, specifically the Mann-Whitney U test, to confirm the robustness of our findings. We have added this clarification to the Methods section

3. Line 331. “Covariates were included in the mixed model if necessary”. Please clarify what covariates were included in the analysis.

Clarified as sugested in the methods section.

4. Line 338. Fat mass, HRmax and VO2 peak were found differently between two groups. In the following analysis, it seems not all three being adjusted in the analysis. What if these were adjusted in the analysis? Were the results and conclusions retained?

Covariates, including HRmax, VO₂ peak, and fat mass, were only included in the mixed linear model when a significant interaction with the primary variables was observed. In cases where no significant interaction was detected, adjustments were deemed unnecessary, as they did not alter the results or conclusions. We confirm that the main findings remain consistent under this approach.

Reviewer #2

In the present study, the authors investigated the acute effects of 3 group exercise classes on arterial stiffness and cardiovascular modulation in both young and middle-aged adults. The manuscript is well-written and organized. However, there are some concerns regarding the transparency of the interventions which must be addressed by the authors. I have provided some comments for authors’ consideration.

Page 9, Line 166; Why the body composition and cardiorespiratory fitness were just measured in the CON session?

Body composition and cardiorespiratory fitness were measured only in the CON session because these variables represent descriptive characteristics of each participant. They were essential for calculating energy expenditure and determining the exercise intensity during the sessions (e.g., as a percentage of VO₂ peak or heart rate reserve). Performing these assessments during the non-exercise condition (CON) ensured that no acute exercise effects (e.g., fatigue, fluid shifts) influenced the results, providing a stable baseline for accurate calculations. Additionally, given that the study duration for each participant was limited to approximately two weeks, we did not expect significant changes in these variables during this period.

Page 9, Line 183; what was the rationale of blinding the participants and fitness instructor until arrival at the lab? Did the individual measuring the outcome measures was blinded regarding the order of exercise intervention?

Participants and the fitness instructor were blinded to the order of the exercise interventions until arrival at the laboratory to minimize any anticipatory psychological effects or behavioral changes that could influence exercise performance or physiological responses. The investigator responsible for outcome measurements was not involved in the delivery of the exercise interventions and remained blinded to the order of the sessions, ensuring unbiased data collection. We have now clarified this aspect to the Methods section.

Did you have any control over the diet 24 or 48 hours before performing each experimental session?

Participants were instructed to maintain their habitual diet and to refrain from alcohol, caffeine, and intense physical activity for at least 24–48 hours before each session. While we did not strictly monitor or control the exact food intake of participants, they were provided with standardized verbal and/or written guidelines to minimize variability in dietary and metabolic states across sessions. We have now added this aspect to the Methods section

Page 10, Line 193; There are some concerns about the details of these 3 interventions. There is missing information on how each training session was personalized for each individual. Typically, the participants have different fitness level and accordingly, each training session must be personalized for them based on their fitness level. How did the author control this in each session. It’s crucial to provide detailed information in this regard. For instance, did a participant aged more than 48 years old perform the same exercise as a 21 years old participant, without considering their age and fitness level?

In this study, the primary aim was to assess the acute cardiovascular and autonomic responses to group exercise classes in an ecologically valid setting, where the sessions were designed to reflect typical gym environments. While the exercises were standardized across all participants, the instructor provided modifications to accommodate individual fitness levels, ensuring participants could adjust the intensity according to their own capabilities. For example, the resistance used in the Pump Power class (PP) was based on individual preferences and goals, allowing participants to choose weights that suited their fitness levels. Similarly, during the BIKE class, participants were encouraged to adjust the cycling cadence and resistance based on their perceived effort, ensuring that each participant could engage in the exercise at an appropriate intensity. Although we did not provide personalized training loads in the strictest sense, this approach ensured that participants, including older and younger individuals, could participate safely and effectively, reflecting the flexibility typical of group fitness classes.

We have now inclued more detailed information on how individual differences in fitness level were addressed within the group fitness classes in the manuscript.

Page 11, Line 210; What if the music performed in each session had a confounding effect on participants? The same music could have different effect on each individual regarding the emotions and motivations.

As this study was designed to reflect an ecologically valid exercise setting, music was intentionally included as a standardized component of the group fitness classes, consistent with real-world practice. To minimize variability, the same music and choreography were used for all participants across sessions. This approach ensured a consistent stimulus while preserving the ecological validity that is central to the study design. A clearer explanation was added to the manuscript.

Page 11, Line 222; What was the duration of each stage in the incremental cycling test? Please provide a reference for this test protocol.

The test followed a continuous ramp design, where the workload increased gradually over time to exhaustion. Specifically, the workload increased continuously by approximately 1 watt every 1.5-3 seconds (equivalent to 20-40 watts per minute), ensuring a smooth and progressive increase in intensity while minimizing premature fatigue. This approach is consistent with established protocols for evaluating maximal aerobic capacity. To address the reviewer’s request, we have added appropriate references for the protocol.

---

## [Decision Letter · Decision Letter 1]

28 Jan 2025

Acute effects of Commercial Group Exercise Classes on Arterial Stiffness and Cardiovagal Modulation in Healthy Young and Middle-aged Adults: a crossover randomized trial

PONE-D-24-44766R1

Dear Dr. Melo,

We’re pleased to inform you that your manuscript has been judged scientifically suitable for publication and will be formally accepted for publication once it meets all outstanding technical requirements.

Kind regards,

Hidetaka Hamasaki

Academic Editor

PLOS ONE

Additional Editor Comments (optional):

Reviewers' comments:

Reviewer's Responses to Questions

**Comments to the Author**

1. If the authors have adequately addressed your comments raised in a previous round of review and you feel that this manuscript is now acceptable for publication, you may indicate that here to bypass the “Comments to the Author” section, enter your conflict of interest statement in the “Confidential to Editor” section, and submit your "Accept" recommendation.

Reviewer #1: All comments have been addressed

Reviewer #2: All comments have been addressed

2. Is the manuscript technically sound, and do the data support the conclusions?

Reviewer #1: (No Response)

Reviewer #2: Yes

3. Has the statistical analysis been performed appropriately and rigorously? 

Reviewer #1: (No Response)

Reviewer #2: Yes

4. Have the authors made all data underlying the findings in their manuscript fully available?

Reviewer #1: (No Response)

Reviewer #2: Yes

5. Is the manuscript presented in an intelligible fashion and written in standard English?

Reviewer #1: (No Response)

Reviewer #2: Yes

6. Review Comments to the Author

Reviewer #1: (No Response)

Reviewer #2: The authors have addressed all my comments properly. I have no further comment. The manuscript is now deemed suitable for publication.

7. PLOS authors have the option to publish the peer review history of their article (what does this mean? ). If published, this will include your full peer review and any attached files.

**Do you want your identity to be public for this peer review?** For information about this choice, including consent withdrawal, please see our Privacy Policy .

Reviewer #1: No

Reviewer #2: No

---

## [Editor Report · Acceptance letter]

PONE-D-24-44766R1

PLOS ONE

Dear Dr. Melo,

I'm pleased to inform you that your manuscript has been deemed suitable for publication in PLOS ONE. Congratulations! Your manuscript is now being handed over to our production team.

Kind regards,

on behalf of

Dr. Hidetaka Hamasaki

Academic Editor

PLOS ONE